# Nanomelanin Potentially Protects the Spleen from Radiotherapy-Associated Damage and Enhances Immunoactivity in Tumor-Bearing Mice

**DOI:** 10.3390/ma12101725

**Published:** 2019-05-27

**Authors:** Nguyen Thi Le Na, Sai Duc Loc, Nguyen Le Minh Tri, Nguyen Thi Bich Loan, Ho Anh Son, Nguyen Linh Toan, Ha Phuong Thu, Hoang Thi My Nhung, Nguyen Lai Thanh, Nguyen Thi Van Anh, Nguyen Dinh Thang

**Affiliations:** 1Faculty of Biology, VNU University of Science, Vietnam National University, Hanoi 100000, Vietnam; betterday2804@gmail.com (N.T.L.N.); saiducloc_t60@hus.edu.vn (S.D.L.); minhtri20081997@gmail.com (N.L.M.T.); nguyenthibichloan.iph@gmail.com (N.T.B.L.); hoangthimynhung@hus.edu.vn (H.T.M.N.); nguyenlaithanh@hus.edu.vn (N.L.T.); 2Department of Pathophysiology, Hoc vien Quan Y, Hanoi 100000, Vietnam; hoanhsonhp@gmail.com (H.A.S.); toannl@vmmu.edu.vn (N.L.T.); 3Institute of Material Sciences, Vietnam Academic of Science and Technology, Hanoi 100000, Vietnam; haphuongthu74@gmail.com; 4National Key Laboratory of Enzyme and Protein Technology, VNU University of Science, Vietnam National University, Hanoi 100000, Vietnam; vananhbiolab@gmail.com

**Keywords:** nanomelanin, radiation protector, X-ray, spleen fibrosis, immunoactivity, cancer treatment

## Abstract

Radiotherapy side-effects present serious problems in cancer treatment. Melanin, a natural polymer with low toxicity, is considered as a potential radio-protector; however, its application as an agent against irradiation during cancer treatment has still received little attention. In this study, nanomelanin particles were prepared, characterized and applied in protecting the spleens of tumor-bearing mice irradiated with X-rays. These nanoparticles had sizes varying in the range of 80–200 nm and contained several important functional groups such as carboxyl (-COO), carbonyl (-C=O) and hydroxyl (-OH) groups on the surfaces. Tumor-bearing mice were treated with nanomelanin at a concentration of 40 mg/kg before irradiating with a single dose of 6.0 Gray of X-ray at a high dose rate (1.0 Gray/min). Impressively, X-ray caused mild splenic fibrosis in 40% of nanomelanin-protected mice, whereas severe fibrosis was observed in 100% of mice treated with X-ray alone. Treatment with nanomelanin also partly rescued the volume and weight of mouse spleens from irradiation through promoting the transcription levels of splenic Interleukin-2 (IL-2) and Tumor Necrosis Factor alpha (TNF-α). More interestingly, splenic T cell and dendritic cell populations were 1.91 and 1.64-fold higher in nanomelanin-treated mice than those in mice which received X-ray alone. Consistently, the percentage of lymphocytes was also significantly greater in blood from nanomelanin-treated mice. In addition, nanomelanin might indirectly induce apoptosis in tumor tissues via activation of TNF-α, Bax, and Caspase-3 genes. In summary, our results demonstrate that nanomelanin protects spleens from X-ray irradiation and consequently enhances immunoactivity in tumor-bearing mice; therefore, we present nanomelanin as a potential protector against damage from radiotherapy in cancer treatment.

## 1. Introduction

Radiotherapy has been applied in cancer treatment for a long time to cure patients by removing the tumor and/or preventing the recurrence of cancer, as well as to reduce physical pain and prolong life [1,2]. However, radiotherapy-mediated killing is not specific to cancer cells and tumors; therefore, it also causes both acute and chronic side effects. There are many examples of side effects, including: fibrosis (the formation of scar tissue), destruction of the bowel, bleeding or diarrhea, impaired memory function or loss of memory, infertility, and the appearance of secondary cancers [2,3]. Radiotherapy also promotes the generation of reactive oxygen species (ROS), which may cause serious DNA damage [4,5]. So far, several drugs have been approved by the Food and Drug Administration (FDA) for clinical use in radiotherapy to protect normal cells and healthy tissues from attack of irradiation, such as amifostine, Neupogen^®^ (filgratism), Neulasta^®^ (pegfilgrastim), and Leukine^®^ (sargramostim); however, the effects of these chemicals are still very limited [6,7,8].

The spleen, as a critical component of the immune system, plays an exclusive role in both the innate and adaptive immune responses [9]. During whole body irradiation, all tissues will be affected, which may result in a reduction of the spleen size and weight, and, consequently, a significant decrease in immune cell numbers [9,10,11]. Immunotherapy, as a method of cancer treatment, currently presents significant potential for the enhancement of populations of immune cells, such as T lymphocytes and dendritic cells (DCs), which play an important role in the suppression of cancer cells [12,13].

Melanin is a natural polymer with low toxicity and has many important roles in the human body, including in the absorption of UV-radiation and as an ROS scavenger [14,15,16]. Most melanin are water-insoluble and just slightly to be solved in Dimethyl sulfoxide (DMSO. Venous injection of melanin as nanoparticles encapsulated into a polymer and/or liposome has been shown to effectively protect bone marrow and blood cells during gamma-radiotherapy [10,17]. However, so far, there have been limited studies focusing on the role of melanin in the protection of normal cells and healthy tissues from irradiation in order to treat solid-tumor cancers. Therefore, in this study, we developed a protocol to make water-dissolving nanoparticles without any aid of polymer or liposome and applied this nanomelanin in radiotherapy to reduce the side effects of X-radiation on healthy tissues of tumor-bearing mice. 

## 2. Materials and Methods

### 2.1. Ethics Statement

All procedures involving animals were conducted following the Principles of Laboratory Animal Care (NIH). The animal experiments in this study were approved by the Ethics Committee from the Military Medical University 103 for Project with code number 08.02-2017.07 funded by National Foundation for Science and Technology Development (NAFOSTED). Approval number: IACUC-025/18.

### 2.2. Preparation of Nanomelanin

Melanin powder (CAS Number: 8049-97-6; Sigma Aldrich, St Louis, MO, USA) was dissolved in sodium hydroxide or ammonium hydroxide solutions at concentrations of 0.1 N, 0.5 N, 1 N, and 3 N to form corresponding nanomelanin solutions. The pH of the nanomelanin solution was then adjusted to pH 7 with hydrochloride acid (HCl) solution. Melanin content was determined according to a previous publication with modifications [18]. Briefly, the content of total melanin was measured by a spectrophotometer at 415 nm using a microplate reader. The morphology and size of nanoparticles was examined using a NANOSEM 450 Scanning Electron Microscope (SEM) (FEI, Hillsboro, OR, USA). Organic and inorganic bonds were characterized in nanomelanin by Fourier-Transform Infrared Spectrophotometry (FTIR) (IRAffinity-1S, SHIMADZU, Kyoto, Japan). A dynamic laser scattering method was applied for the measurement of surface potential and size distribution using a Zetasizer (software v7.11, Malvern Instruments, Worcestershire, UK).

### 2.3. Cells and Mice

Murine lewis lung cancer 3LL cells (ATCC CRL-1642) were cultured in RPMI-1640 supplemented with 10% fetal bovine serum (FBS) and 1% penicillin/streptomycin at 37 °C in 5% CO_2_. Five-week-old Swiss mice, purchased from the National Institute of Hygiene and Epidemiology (NIHE; Hanoi, Vietnam), were used in this study. Mice were maintained in cages at a temperature of 24 ± 3 °C and 55% humidity and were fed twice daily.

### 2.4. Realtime-Polymerase Chain Reaction (PCR) Analysis

Total RNA was isolated from mouse tissue using GeneJET RNA Purification (Thermo Scientific, Waltham, MA, USA) according to the manufacturer’s instructions. Then cDNA was synthesized from total-RNA by reverse transcriptase reaction, according to the protocol supplied with the RevertAid First Strand cDNA Synthesis Kit (Thermo Scientific, Singapore). Real-time quantitative reverse transcription polymerase chain reaction (RT-PCR) was performed in a LightCycler® 96 Instrument (Roche Diagnostics GmbH, Mannheim, Germany). Expression levels of TNF-α, IL-2, Bax, and caspase-3 transcripts measured by real-time quantitative RT-PCR were adjusted through the transcript expression level of β-actin (for tumor samples) or Glyceraldehyde-3-Phosphate Dehydrogenase (GAPDH) (for spleen samples). PCR was carried out using 10 μL power SYBR1 Green PCR master mix, with 900 nM each of forward and reverse primers in a final volume of 20 μL. All primer sequences are presented in Table 1.

### 2.5. Tumor-Bearing Mice

Murine lewis lung cancer 3LL cells, at a concentration of 5 × 10^6^ cells/150 μL of Phosphate-Buffered Saline (PBS), were injected subcutaneously under the skin of the back. Formed tumors were observed and measured every 3 days. Approximately 10 days post injection with 3LL cells, when the tumors reached the appropriate size of approximately 500 mm^3^ (10 × 10 × 5 mm^3^), mice were treated with nanomelanin and exposed to X-ray irradiation.

### 2.6. Radiotherapy Treatment of Tumor-Bearing Mice

Mice were allocated to four groups (n = 5, each) as follows: Group 1 (NIL), no tumor induction and no nanomelanin treatment or X-ray radiation; Group 2 (NC), tumor-bearing mice without nanomelanin treatment or X-ray radiation; Group 3 (IR), tumor-bearing mice treated with X-ray radiation only; and Group 4 (IR + MEL), tumor-bearing mice treated with both nanomelanin and X-ray radiation. Previously, tumor-bearing mice were treated with/without melanin; however, no difference was observed between these two groups. Therefore, the control group of tumor-bearing mice treated with nanomelanin only was omitted in this report. Tumor-bearing mice were treated with a single dose of 6.0 Gy with a high dose rate (HDR) of 1.0 Gy/min using a Precise Digital Accelerator 152377 (ELEKTA, Stockholm, Sweden) at Military Hospital 103. Mice in group 3 received a single dose of X-ray radiation. Mice in group 4 also received abdominal injection of 40 mg/kg nanomelanin 4 h before the single dose of X-ray radiation and again 48 h post-irradiation. Mice were sacrificed 15 days post-irradiation.

### 2.7. Flow Cytometry Analysis

Spleens were immediately collected following sacrifice and washed with PBS three times. Spleen samples (50 g) were placed in 2 mL of PBS and gently ground using a glass rod to generate a homogeneous solution, then the samples were filtrated through a 70 μm pore-size membrane. The filtrate was then centrifuged at 1500 rpm for 5 min at 4 °C. The pellet was resuspendend in 1.5 mL of Erythrocyte Lysis Buffer (ELB) (BD Biosciences, US) and kept at room temperature for 10 min. The suspension was centrifuged again under the same conditions, and the resulting pellet was washed with PBS three times. The cell pellet was then resuspended in 3 mL of FACS buffer and kept at 4 °C. Then, 10^6^ of this solution was added to 100 µL of a diluted antibody in FACS buffer and incubated for 1 h (anti-CD19-PE for B cells, anti-CD11c-FITC for dendritic cells, and anti-CD3-PE for T cells; BD Biosciences, Franklin Lakes, NJ, USA). The mixture was then gently washed with FACS buffer three times before measuring on a BD FACS Lyse flow cytometer.

### 2.8. Histological Analysis

Spleen and tumor tissues were collected and kept in formalin (10%). Tissues were then eliminated of calcium using formic acid (5%) and fixed in paraffin. Next, paraffin-embedded tissues were sectioned and fixed onto microscope slides. Formalin-fixed paraffin-embedded slides were deparaffinized in xylene and subsequently dehydrated with alcohol before staining with hematoxylin and eosin.

### 2.9. Statistical Analysis 

The SPSS (version 18) software package (SPSS Inc., Chicago, IL, USA) was used for statistical analyses, and the significance level was set at p < 0.05. Results were statistically analyzed using a Student’s *t*-test.

## 3. Results and Discussion

### 3.1. Preparation and Characterization of Nanomelanin from Melanin Powder

Melanin has a polymeric structure and very low water-solubility. The structure and chemical properties on the surface of the melanin powder were examined and presented in Figure 1. This analysis revealed that melanin powder consists of a multi-particle structure (Figure 1A,B) with the presence of various functional groups, including hydroxyl (-OH), carbonyl (-C=O), and carboxyl (-COO) groups (Figure 1C) [19,20,21]. In general, to increase the water-solubility of melanin, nanomelanin particles are prepared by wrapping into liposomes or polymers [10,17]. Previous studies have also revealed that melanin as a nanoparticle has a much stronger capacity for absorption of UV-rays to protect DNA from damage [22,23].

In this study, nanomelanin particles were produced without the aid of liposomes or polymers. The optimal conditions for dissolving melanin were examined using different concentrations of NaOH (Figure 2A–D) and NH_4_OH (Figure 2E–H). Melanin was degraded into smaller particles in solutions of NH_4_OH as follows: 0.1 N, 625 nm; 0.5 N, 260 nm; 1 N, 294 and 112 nm; and 3 N, 2685 and 354 nm. In solutions of NaOH, particle sizes were as follows: 0.1 N, 204 and 64 nm; 0.5 N, 160 nm; 1 N, 4006 and 340 nm; and 3 N, 681, 9073, and 5580 nm (Figure 2). Melanin contents in NaOH 0.1 N, 0.5 N, 1 N, and 3 N were 0.73 mg/mL, 2.21 mg/mL, 2.86 mg/mL and 3.65 mg/mL, respectively; while melanin contents in NH_4_OH 0.1 N, 0.5 N, 1 N and 3 N were 0.21 mg/mL, 0.84 mg/mL, 1.14 mg/mL and 2.03 mg/mL, respectively. It indicated that NaOH was better than NH_4_OH at dissolving melanin powder. In addition, it also demonstrated that the stronger the alkali condition, the higher the dissolvability of melanin. However, at high concentrations of NaOH (1 N and 3 N), melanin was broken down into many different fragments (Figure 2C,D).

Based on these results, the solution of 0.5 N NaOH was chosen as the best condition for breaking down the melanin powder into nano-scale particles. Under these conditions, nanomelanin size ranged from 80–200 nm, with an average size of 160 nm (Figure 2B). Further, the nanomelanin particles (in 0.5 N NaOH) were observed under SEM at magnifications of 25,000 × (Figure 3A) and 50,000 × (Figure 3B). It has been proven that the existing form of melanin plays an important role in the absorption of radiation. The chemical properties of nanomelanin were also examined using FTIR spectrometry and presented in Figure 3C. Consistent with the findings of the prior analysis of the composition of melanin powder in Figure 1, several functional groups were identified, including -OH, at around 3300 cm^−1^; -C=O), at around 1630 cm^−1^; and -COO, at around 1045 cm^−1^ (Figure 3C).

### 3.2. Nanomelanin Protected Mouse Spleens from X-radiation

First, the toxicity of nanomelanin was tested. Mice were injected with two shots (2-day interval) of 1 mL nanomelanin at different concentrations of 5, 10, 20, 40, 60, and 80 mg/kg. It showed that, at the maximum concentration of 60 mg/kg, nanomelanin did not cause any defect or strange symptoms for mice. However, at high concentrations of 80 mg/kg, although mice were still alive they somehow lost appetite and decreased movement. In addition, previous studies also reported that nanomelanin at the concentration of 50 mg/kg was safe for mice [10,11]. Based on these results, we decided to choose a safety dose of 40 mg/kg (but high enough) for the radioprotecting experiments. Mice were first irradiated with different doses of 4, 5, 6, and 7 Gy of X-ray with a high dose rate of 1 Gy/min. The obtained results indicated that at the doses of 4, 5 and 6 Gy, the X-ray did not cause any death of mice after 20 days post-irradiation; however, the dose of 7 Gy caused some deaths of mice after 7 days post-irradiation. This result was consistent with results reported in previous studies [24,25,26,27]. Based on this, we selected the dose of 6.0 Gy for further experiments. Tumor size and mouse weight were measured every 3 days. When tumor size reached a volume of approximately 500 mm^3^ (10 × 10 × 5 mm) (Figure 4A–C) on day 10, mice were treated with nanomelanin and irradiated with a single dose X-ray of 6.0 Gy; mice were sacrificed 15 days after irradiation. It was clear that irradiation severely retarded the weight development of mice (Figure 4D). Although the weight of mice in both the IR and IR + MEL groups was significantly lower than that in the NC and NIL groups, weight in the IR + MEL group was higher than that in the IR group (Figure 4D). In addition, radiation also halted and then decreased the development of tumors in mice in the IR and IR + MEL groups. Furthermore, the trend in decreasing tumor size was more obvious in the IR + MEL group than that in the IR group (Figure 4E), although this difference was not significant.

In addition, hematological analyses revealed that irradiation resulted in a serious reduction in white blood cells (WBC), hematocrit (HCT), and platelets (PLT), and a slight decrease in red blood cell (RBC) and hemoglobin (HGB) with significant differences (Table 2).

Although WBC was lower in irradiated mice, the proportion of lymphocytes (LYM) was higher than that observed in the NC group. More importantly, WBC, PLT, and LYM were significantly higher in mice in the IR + MEL group than in mice in the IR group (Table 2). Although irradiation resulted in a substantial reduction in spleen size and weight (Figure 5A), the weight of spleens from mice in the IR + MEL group were significantly higher than those of mice in the IR group (P < 0.05) (Figure 5B). Further, histological analyses were performed to investigate any changes in organization of mouse spleens. The spleen of mice in the NIL and NC groups appeared normal, which is indicated by the presence of a large amount of evenly-distributed white pulp containing mononuclear cells (Figure 5C,D). All the spleen of mice in the IR group (5/5) demonstrated severe fibrosis (Figure 5E), while fibrosis was only observed in 40% (2/5) of spleens from mice in the IR + MEL group and was mild in these cases (Figure 5F). Evidence of fibrosis was indicated by the presence of fibrous cells with bizarre nuclei (multinuclear cells with sporadic distribution), especially in the reticuloendothelial system (Figure 5E,F red frame). Moreover, microscopic evaluation revealed that the numbers of areas of white pulp were much reduced in the IR group compared to the IR + MEL group (Figure 5E,F, red arrows). Previous studies have reported that the activation of IL-2 plays an important role in increasing splenic weight and that TNF-α is a pro-inflammatory cytokine, inducing a broad range of cellular responses, ranging from inflammatory cytokine production, cell survival, cell proliferation, and cell differentiation [28,29,30]. Therefore, here we analyzed the expression levels of IL-2 and TNF-α extracted from spleen tissue by RT-PCR. Although IL-2 (Figure 6A) and TNF-α (Figure 6B) expressions were inhibited in both the IR and IR + MEL groups compared to the NC group, expression levels were higher in the IR + MEL group than in the IR group by 1.72 and 2.14-fold, respectively (Figure 6).

### 3.3. Nanomelanin Enhanced Populations of Immune Cells in the Spleen of X-ray Irradiated-Mice

Spleen is responsible for the production of immune cells, and therefore, the changes in splenic immune cell populations were investigated as follows: B cells, CD19 marker; T cells, CD3 marker; and DCs, CD11c marker. While CD3 and CD19 are very specific markers for T cells and B cells, respectively, although CD11c presents mainly on the surfaces of DCs and is widely accepted and used as the most important marker for DCs, it may also appear on the surfaces of some other cells in certain circumstances [31,32,33,34,35]. In the spleen, besides B cells, T cells and DCs, there are several other cells such as monocytes, macrophases, and neutrophils. In normal conditions, monocytes, macrophases, and neutrophils are CD11c-negative cells [31,32,33,34]; however, when spleen inflammation is induced by the infection of bacteria and/or bacterial derivatives, a small amount of monocytes, macrophases, and lymphocytes will be positive with CD11c [33,34,35]. However, in this study, mice were not infected with bacteria and/or bacterial derivatives; therefore, almost all CD11c-positive cells in the spleens of mice should be DCs. Irradiation resulted in a strong reduction of B cells in both IR and IR + MEL groups (Figure 7A–E). In contrast, there were slight, yet significant increases in the populations of T cells (Figure 7F–J) and dendritic cells (Figure 7K–P) in IR and IR + MEL groups, respectively. Specifically, the populations of T cells and dendritic cells were 1.91 and 1.86-fold higher, respectively, in the IR + MEL group than those in the IR group (Figure 7E,J). This increase in number of immune cells may relate to the better splenic conditions observed in mice in the IR + MEL group compared to those in mice in the IR group.

### 3.4. Nanomelanin Indirectly Activated Apoptotic Signaling in the Tumor Tissues of Mice

Histological analysis was conducted to confirm the presence of tumors in the mice in the NC group (Figure 8A,B), IR group (Figure 8C,D) and IR + MEL group (Figure 8E,F).

It was clear that the tumor consisted of epithelial lung cancer cells with irregular, hyperpigmented and multiple nuclei. There were necrotic nodes located in the tumors; however, there was no morphological difference, with the exception of size, between tumors in non-irradiated and irradiated-mice. Besides the expression as a radioprotector, melanin was reported as a target molecule for cancer treatment, especially for melanoma treatment [36,37,38,39]. Because melanin is synthesized and released by melanoma cells, therefore, it could be used as a target molecule for photothermal therapy and/or radioisotope therapy melanoma treatment rather than for the treatment of other cancers [36,37,38,39]. In this study, we used lung cancer cells for the experiments; therefore, it was hard to see the impressive effect of melanin as an anticancer agent. However, the expression levels of several genes which play important roles in apoptotic signaling in tumor tissues, such as TNF-α, Bax, and Caspase-3, were examined. TNF-α expression in tumor tissues of mice in the IR group was only one-third of the level detected in tumor tissues of mice in the NC-group. This reduction in TNF-α may be due to splenic damage (reduced size and heavy fibrosis), as this is typically where TNF-α is produced. Of note, TNF-α expression was significantly higher (1.84-fold) in the IR + MEL group than that in the IR group (Figure 8G). Moreover, Bax and Caspase-3 expression levels were also significantly induced in the tumor tissues of the IR + MEL group compared to those in the IR group (1.62 and 1.41-fold higher, respectively) (Figure 8H,I).

## 4. Discussion

Radiotherapy is commonly used in cancer treatment; however, it may cause severe side effects; therefore, development of radiation protecting drugs (RPDs) has been a focal point for a long time. RPDs used to reduce the side effects associated with radiotherapy can be classified into three groups: radioprotectors, radiation mitigators, and therapeutic agents. Radioprotectors protect cells from damage during irradiation by enhancing antioxidant activity to eliminate free radicals [6]; radiation mitigators are normally administered after irradiation, but before the appearance of side-effect symptoms, to promote DNA repair and activate redox systems; therapeutic agents are administered after the appearance of side effect symptoms [7]. To date, RPDs from each of these categories have been approved by the FDA for clinical use, such as amifostine, filgratism, pegfilgrastim, and sargramostim [8]. Following this classification, melanin is a radioprotector because of its properties in absorbing radiation and eliminating ROS generated during and after radiotherapy. In this study, nanomelanin was administered just before and 2 days post-X-ray irradiation to increase the ability of melanin to protect normal cells and healthy tissues.

Melanin is an insoluble polymer with many functional groups, including hydroxyl, carbonyl, and carboxyl. Previous studies reported that melanin as a nanoparticle, rather than dissolved-melanin, was highly effective in protecting DNA from attack by UV radiation [19,23]. In several previous studies, nanomelanin as polimer and/or liposome-coated forms was used as a protecting agent in certain irradiation experiments [10,11]. In this study, nanomelanin was made by a novel method by breaking down melanin powder under alkaline conditions to form nano-scale particles in the size range of 80–200 nm, without their aid of either a polymer and/or liposome (Figure 2B and Figure 3A,B). The formation of nano-scale particles may increase beneficial properties, including solubility, effective distribution, and efficacy in absorption of radiation. In addition, FTIR analysis confirmed that there were abundant functional groups including hydroxyl, carbonyl and carboxyl groups, which may provide excellent antioxidant activity for melanin to scavenge ROS generated during and after X-ray irradiation (Figure 3C). These characteristics of nanomelanin may have contributed to its partial protection of mouse spleens from the attack of the X-ray radiation.

Currently, immunotherapy is one of the newest and most promising methods for cancer treatment [9]. In this study, we showed that nanomelanin protected the spleen, which plays a key role in both the innate and adaptive immune systems, from attack by X-ray irradiation, by significantly enhancing splenic weight and reducing fibrosis (Figure 5) via upregulation of IL-2 and TNF-α expression (Figure 6). Previous studies have reported that volume/weight-loss and fibrosis symptoms are frequently observed in the spleen and liver tissues during radiotherapy [11,12,13,40]. It has also been demonstrated that the activation of IL-2 led to an increase in weight and volume of spleens [28]. TNF-α, a pro-inflammatory cytokine, regulates many cellular responses including further inflammatory cytokine production; cell survival, proliferation, and differentiation; and cell apoptosis and necrosis [29,30]. The increase of TNF-α expression in the spleens of mice in the IR + MEL group may relate to survival and proliferation of splenic cells and subsequent enlargement of spleens after irradiation. These results suggested that nanomelanin likely protected the spleens of these mice by reducing the effect of X-ray irradiation and partly rescuing IL-2 and TNF-α expression, which were otherwise suppressed during irradiation. Further, our results revealed that although X-ray irradiation resulted in a marked decrease of B cells (Figure 7A–E), it led to a slight increase of T cells (Figure 7F–J) and DCs (Figure 7K–P) in the mouse’s spleens. More interestingly, treatment with nanomelanin strongly promoted the populations of T cells (Figure 7F–J) and DCs (Figure 7K–P) in spleens of X-ray irradiated-mice. These results are consistent with the data on lymphocyte (LYM) levels in blood, presented in Table 2. A previous study reported that under certain conditions radiation can enhance the immune response of organisms [41]. It has also been reported that radiation exposure during cancer treatment can provide a source of antigen that is well-suited for cross presentation by the dendritic cells and T cells, and further activation of T cells by dendritic cells [6,7,42,43,44,45]. Further, the role of IL-2 in enhancing the population of tumor-reactive T cells was also reported [46]. However, in this study, we, for the first time, indicated the possibility of a synergistic effect of X-ray and nanomelanin in promoting the population of T-lymphocytes and dendritic cells in spleens of mice. In addition, this study also revealed that the intrinsic apoptosis pathway (TNF-α/Bax/Caspase-3) activation was enhanced in the tumor tissues of mice in the IR + MEL group compared to the IR group (Figure 8). These results suggest that the increase in T cell and dendritic cell populations in spleens may help to promote the apoptotic pathway in tumor tissues via upregulation of TNF-α, Bax, and Caspase-3 expression. This data is consistent with a previous report on the promotion of immune cells to kill cancer cells that circulate in the blood and/or in the tumor tissues via apoptosis and/or necrosis pathways [47].

## 5. Conclusions

In this study, we prepared and characterized nanomelanin and investigated its ability in protecting healthy tissues of tumor-bearing mice from the attack of X-ray during radiotherapy. Our results demonstrated that nanomelanin could effectively protect the spleens of mice from X-ray irradiation by reducing fibrosis, enhancing volume/weight of spleen, and rescuing the transcript levels of splenic IL-2 and TNF-α, which were affected by radiotherapy. Further, our results revealed that treatment with nanomelanin strongly promoted the populations of T cells and dendritic cells in spleens of X-ray-irradiated mice. Taken together, we suggest that nanomelanin is a promising radioprotector and, therefore, could be applied in radiotherapy to protect the spleen from radiation during the treatment of cancer.

## Figures and Tables

**Figure 1 materials-12-01725-f001:**
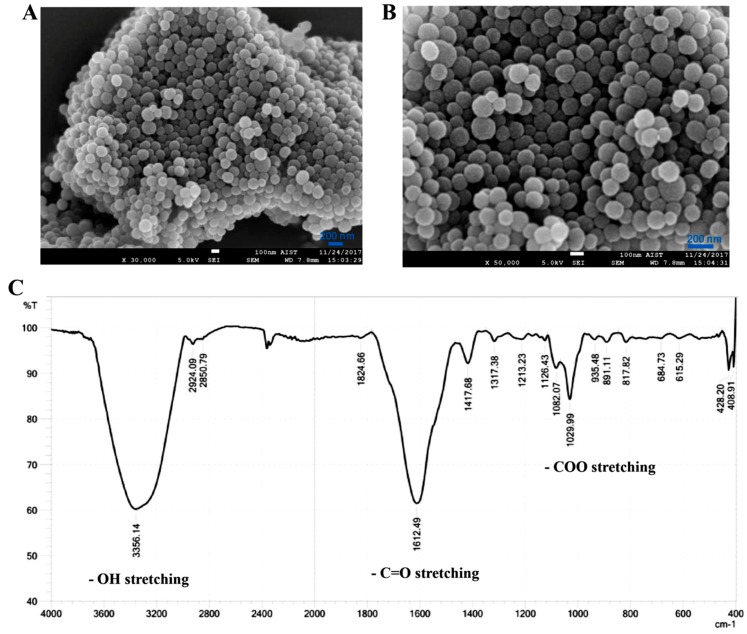
Melanin powder under a SEM at different magnifications of 30,000 × (**A**) and 50,000 × (**B**); and chemical functional groups detected on melanin, measured by FTIR (**C**).

**Figure 2 materials-12-01725-f002:**
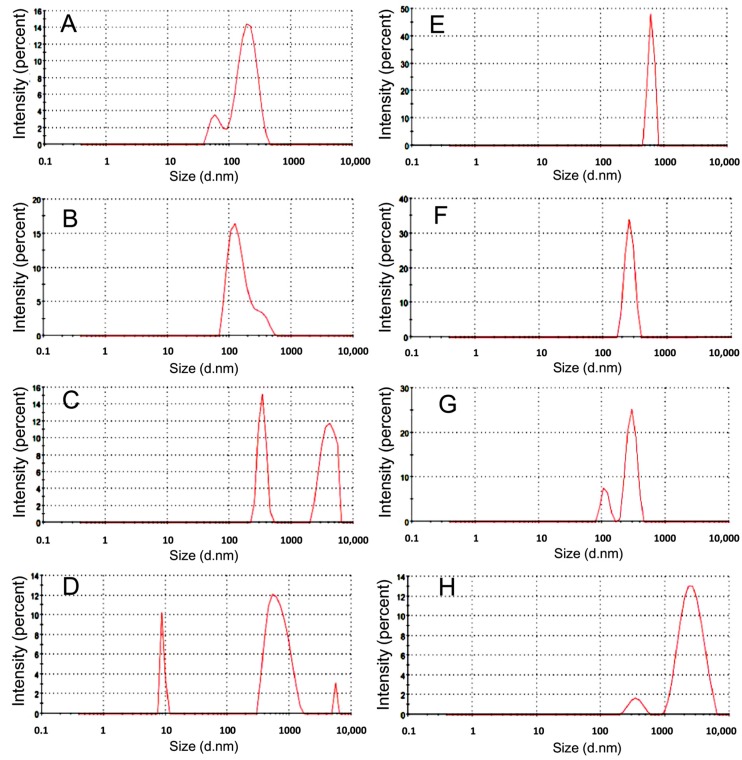
Melanin particle distributions in NaOH solutions at concentrations of 0.1 N (**A**), 0.5 N (**B**), 1 N (**C**), and 3 N (**D**), and in NH_4_OH solutions at concentrations of 0.1 N (**E**), 0.5 N (**F**), 1 N (**G**), and 3 N (**H**); determined by dynamic light scattering, using Zetasizer technology.

**Figure 3 materials-12-01725-f003:**
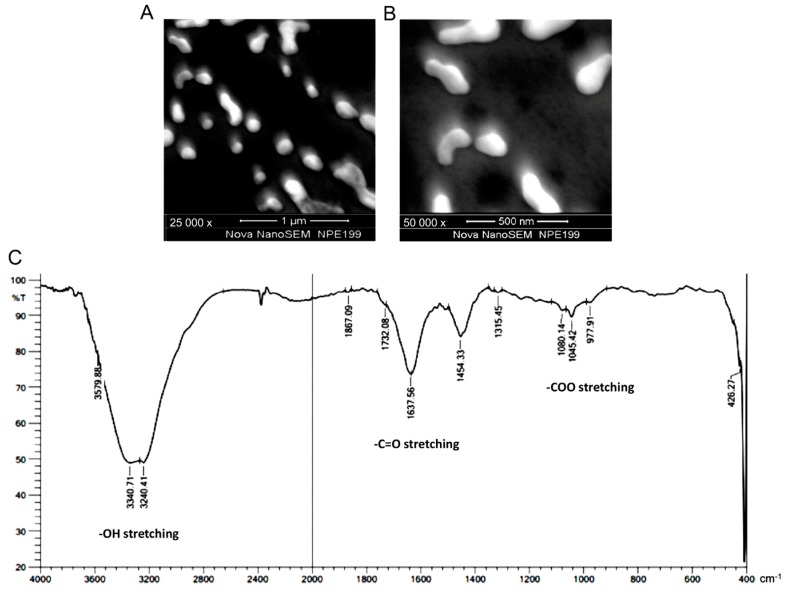
Nanomelanin particles under SEM at different magnifications of 25,000 × (**A**) and 50,000 × (**B**); and chemical functional groups detected on nanomelanin particles, measured by FTIR (**C**).

**Figure 4 materials-12-01725-f004:**
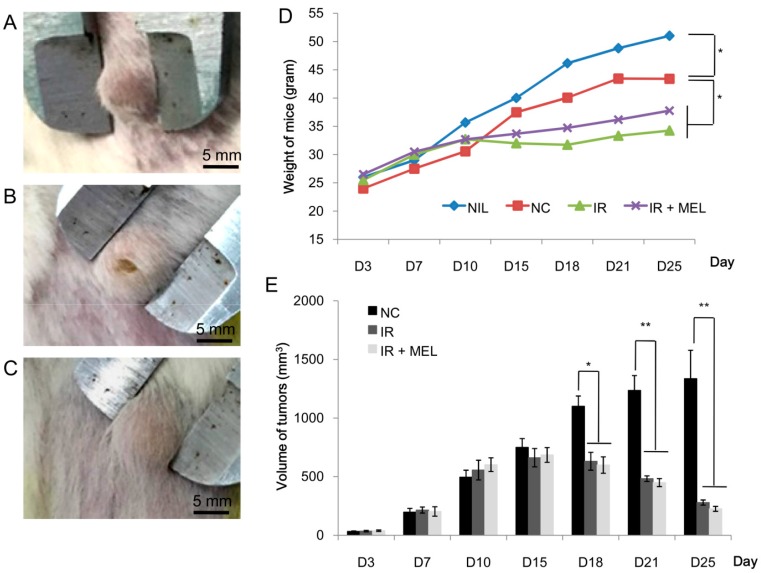
Tumor formation on the back of the NC control mouse (**A**), IR mouse (**B**) and IR + MEL mouse (**C**); weight of mice (g) over time (**D**); development of tumors on the backs of mice over time (**E**); NIL, mice with no induction of tumor or treatment; NC, mice receiving no treatment; IR, mice treated with a single dose of X-ray radiation; IR + MEL, mice treated with 40 mg/kg nanomelanin once before X-ray radiation and again 2 days post-radiation. *, **: Significant differences with P < 0.05 and P < 0.01, respectively.

**Figure 5 materials-12-01725-f005:**
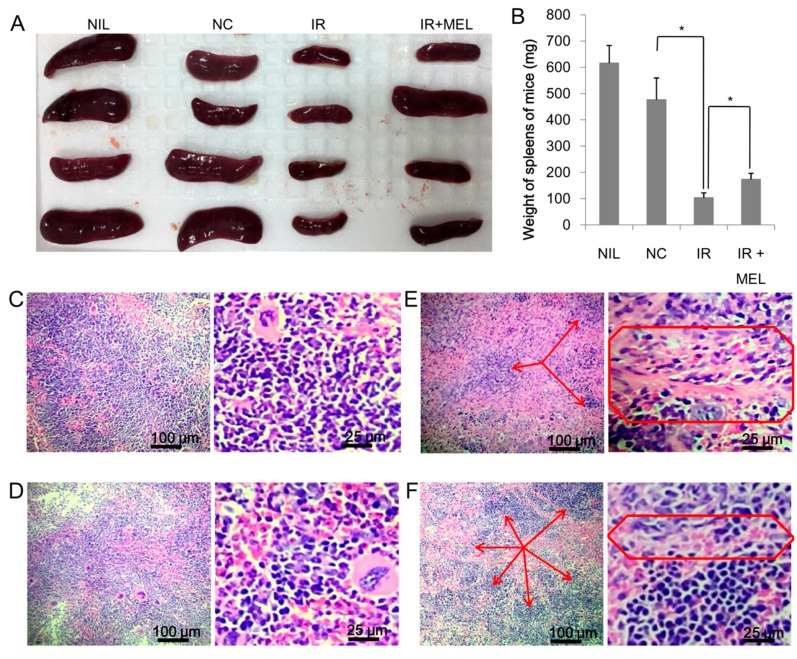
Spleen samples of mice in NIL, NC, IR and IR + MEL groups (**A**); weight of spleens of mice in each treatment group (**B**). Histological organization of spleens from mice in the NIL (**C**), NC (**D**), IR (**E**), and IR + MEL (**F**) treatment groups were observed under a microscope at two different magnifications of 1× (left panels) and 4× (right panels). The appearance of white pulp (arrows) and the reticuloendothelial system (boxed frame) with the presence of fibrous cells. NIL, mice with no induction of tumor or treatment; NC, mice receiving no treatment; IR, mice treated with a single dose of X-ray radiation; IR + MEL, mice treated with 40 mg/kg nanomelanin once before X-ray radiation and again 2 days post-radiation. *: Significant differences with P < 0.05.

**Figure 6 materials-12-01725-f006:**
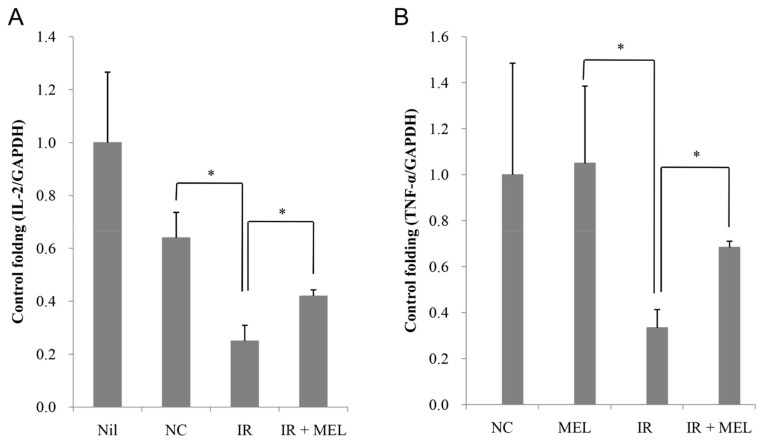
Expression levels of IL-2 (**A**) and TNF-α (**B**) in the spleen of mice in the NIL, NC, IR and IR + MEL treatment groups. NIL, mice with no induction of tumor or treatment; NC, mice receiving no treatment; IR, mice treated with a single dose of X-ray radiation; IR + MEL, mice treated with 40 mg/kg nanomelanin once before X-ray radiation and again 2 days post-radiation. *: Significant difference with P < 0.05 from the control by the Student’s *t*-test.

**Figure 7 materials-12-01725-f007:**
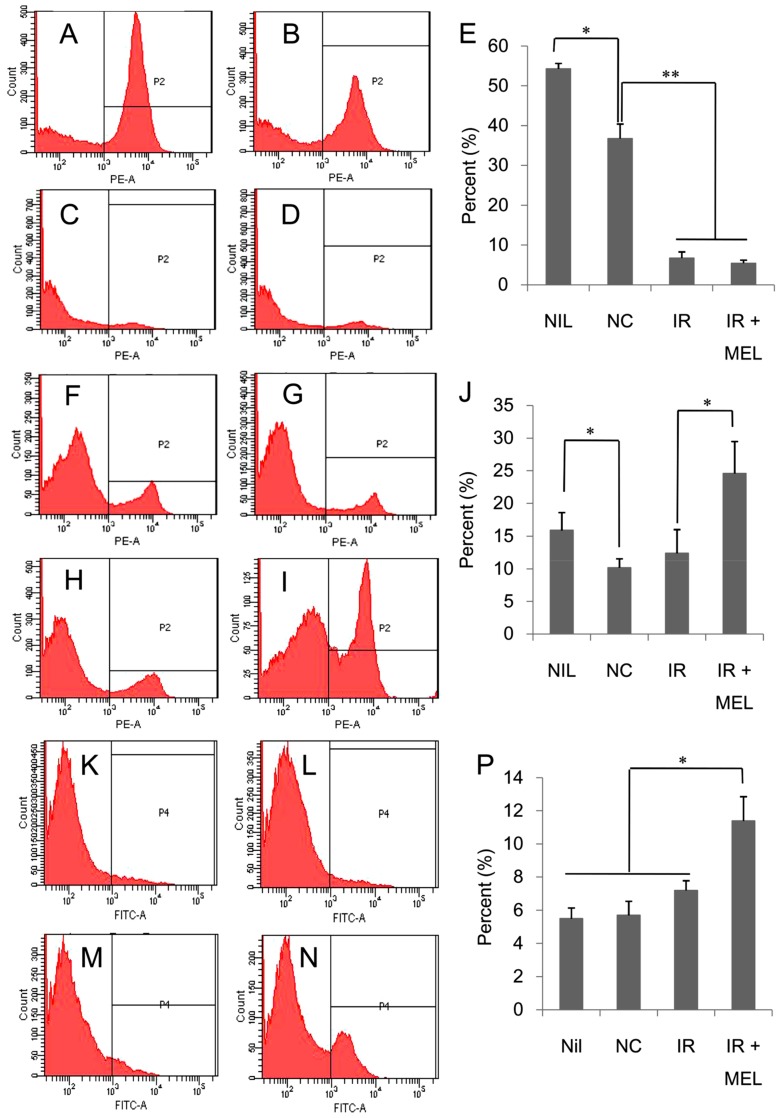
Populations of B cells in spleens of mice in the NIL (**A**), NC (**B**), IR (**C**), and IR + MEL (**D**) treatment groups determined by flow cytometry. Percentages of populations of B cells in each group are presented as histograms (**E**). Populations of T cells in spleens of mice in NIL (**F**), NC (**G**), IR (**H**), and IR + MEL (**I**) treatment groups determined by flow cytometry. Percentages of populations of T cells in each group are presented as histograms (**J**). Populations of dendritic cells in spleens of mice in NIL (**K**), NC (**L**), IR (**M**), and IR + MEL (**N**) treatment groups determined by flow cytometry. Percentages of populations of DC cells in each group are presented as histograms (**P**). NIL, mice with no induction of tumor or treatment; NC, mice receiving no treatment; IR, mice treated with a single dose of X-ray radiation; IR + MEL, mice treated with 40 mg/kg nanomelanin once before X-ray radiation and again 2 days post-radiation. *, **: significant differences with P < 0.05 and P < 0.01, respectively, from the control by the Student’s *t*-test.

**Figure 8 materials-12-01725-f008:**
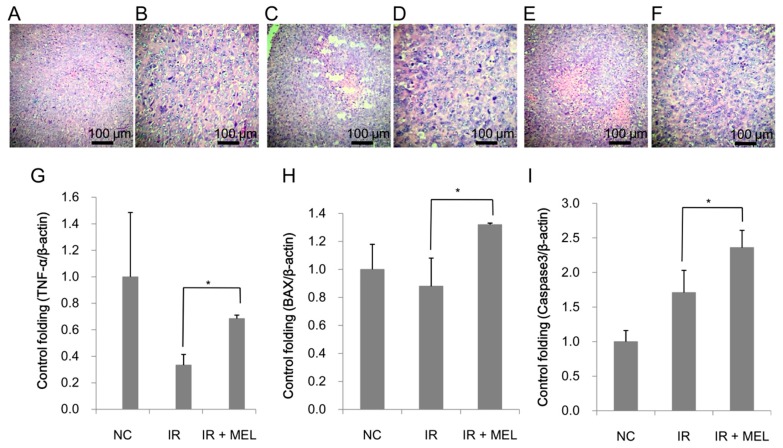
Histological organization of tumor tissues of mice in NC (**A**,**B**), IR (**C**,**D**) and IR + MEL (**E**,**F**) treatment groups. The relative expression levels of TNF-α (**G**), Bax (**H**), and Caspase 3 (**I**) in tumor tissues of mice in NC, IR, and IR + MEL treatment groups. β-actin was used as an internal control. NC, mice receiving no treatment; IR, mice treated with a single dose of X-ray radiation; IR + MEL, mice treated with 40 mg/kg nanomelanin once before X-ray radiation and again 2 days post-radiation. *: Significant difference with P < 0.05 from the control by the Student’s *t*-test.

**Table 1 materials-12-01725-t001:** Primer sequences for genes.

Gene Name	Sequence (5′ to 3′)
mouse TNF-α (F)	ATGAGCACAGAAAGCATGA
mouse TNF-α (R)	AGTAGACAGAAGAGCGTGGT
mouse IL-2 (F)	TTGTGCTCCTTGTCAACAGC
mouse IL-2 (R)	CTGGGGAGTTTCAGGTTCCT
mouse Caspase-3 (F)	CCTCAGAGAGACATTCATGG
mouse Caspase-3 (R)	GCAGTAGTCGCCTCTGAAGA
mouse Bax (F)	AGCAAACTGGTGCTCAAGGC
mouse Bax (R)	CCACAAAGATGGTCACTGTC
mouse GAPDH (F)	CCCATCACCATCTTCCAGGAGC
mouse GAPDH (R)	CCAGTGAGCTTCCCGTTCAGC
mouse β-actin (F)	CGGTTCCGATGCCCTGAGGCTCTT
mouse β-actin (R)	CGTCACACTTCATGATGGAATTGA

**Table 2 materials-12-01725-t002:** Lab results for hematological analysis.

Parameters	Unit	NIL	NC	IR	IR + MEL
White Blood Cell (WBC)	[10^9^/L]	4.15 ± 0.98	3.8 ± 0.94	0.45 ±0.10	0.7 ± 0.22
Red Blood Cell (RBC)	[10^12^/L]	7.7 ± 0.52	6.68 ± 0.19	5.48 ± 1.09	5.21 ± 0.68
Hemoglobin (HGB)	[g/dL]	11 ± 0.61	10.63 ± 0.93	8.34 ± 0.65	7.77 ± 0.69
Hemocratit (HCT)	[%]	34.4 ± 2.69	33.1 ± 2.30	24.94 ± 2.05	23.1 ± 1.21
Platelets (PLT)	[pg]	763.5 ± 27.4	801.6 ± 52.8	178.14 ± 62.5	218.7 ± 23.0
LYM%	[%]	72.22 ± 1.52	37.83 ± 2.80	55.96 ± 3.82	60.9 ± 3.15

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
