# Peer review of "Nanomelanin Potentially Protects the Spleen from Radiotherapy-Associated Damage and Enhances Immunoactivity in Tumor-Bearing Mice"

_materials, 2019, doi:10.3390/ma12101725_

Round 1

Reviewer 1 Report

The authors have synthesized Melanin nanoparticles and showcased their application as radioprotective agents. The authors have done a number of characterizations and in vivo studies. Also, the topic is very interesting and well-suited for Materials journal. I recommend publication of this manuscript after addressing the following major and minor issues.

Major issues:

The authors have used 4 groups to evaluate the effect of the Melanin nanoparticles on the spleen of mice. One control group seems to be missing: Group 5: Tumor-bearing mice treated with Melanin nanoparticles only. Melanin nanoparticles are considered as iron chelation agents and can destroy cancer cells by iron starvation (10.1007/s10856-018-6190-x). Therefore, the combination of X-ray radiation and Melanin nanoparticles may consider as both radiotherapy and targeted drug therapy for cancer treatment, and the Melanin nanoparticles may not necessarily reduce the side effect of radiotherapy but they may directly kill cancer cells. The authors need to discuss this issue. In addition, the novelty of the work is not clear. Is it the effect of Melanin nanoparticle on spleen or is the way that they have synthesized these nanoparticles? They need to elaborate on this matter in the introduction part. In Conclusion, the authors have mentioned: “In this study, we for the first time, prepared and characterized nanomelanin and investigated its ability in protecting healthy tissues of tumor-bearing mice from attack of X-ray during radiotherapy”. However, Rageh et al. in 2014 had already evaluated the effect of Melanin nanoparticles on spleen (10.1007/s11010-014-2232-y). Furthermore, the authors have only used a single concentration (40 mg/kg) of Melanin nanoparticles as well as a single dose of radiation (6.0 Gy)  for their in vivo studies. Why have they selected this specific concentration and radiation dose? It would be more desirable to check the effect of various concentrations of melanin nanoparticles as well as radiation intensity and exposure time.

 Minor issues:

1)    It’s better to add the size and concentration of nanoparticle in the abstract.

2)    Figure 1 A-B needs scale bars.

3)    The titles of vertical axes in Figure 4 B-C & Figure 5B are missing (there is only the unit of the axis)

4)    The authors have mentioned that on day 15 they sacrificed the mice. So, what are D18, D21 & D25 on Figure 4 B-C?

5)    There are several acronyms that have not been defined for the first time such as:

-Line 29: IL-2 (which is defined in Line 236), TNF-α

- Line 67: DMSO

- Line 100: PCR

- Line 104: RT

       - Line 114: PBS

Also, some acronyms such as SEM and FTIR have been defined several times!

6)    There are several syntax and grammatical errors, including, but not limited to, Lines 224, 236, 243, 370. The authors need to proofread the manuscript carefully.

Author Response

Dear Reviewer,

First of all, we would like to thank Editors and Reviewers for their considerations and reasonable comments to our manuscript. We have followed all the comments to revise our manuscript by making clear several points, supplying additional data and correcting all the typo and grammar mistakes in the manuscript.

Now we would like to re-submit our revised-manuscript entitled “Nanomelanin potentially protects the spleen from radiotherapy-associated damage and enhances immunoactivity in tumor-bearing mice”, (manuscript ID: materials-503306) to Materials of MDPI for your further considerations. In the revised manuscript, the revised parts are maked with red color.

In this letter, we also attach our responses to the comments of the Reviewer.

Thank you so much.

Best regards,

Corresponding author: Nguyen Dinh Thang

²  Response to the Reviewer 1

The authors have synthesized Melanin nanoparticles and showcased their application as radioprotective agents. The authors have done a number of characterizations and in vivo studies. Also, the topic is very interesting and well-suited for Materials journal. I recommend publication of this manuscript after addressing the following major and minor issues.

Major issues:

The authors have used 4 groups to evaluate the effect of the Melanin nanoparticles on the spleen of mice. One control group seems to be missing: Group 5: Tumor-bearing mice treated with Melanin nanoparticles only.

è We totally agree with the Reviewer that we may miss a control group. However, actually, before concentrating on investigating the role of nanomelanin as a radioprotector, we had also tried to examine the effect of melanin on cancer treatment. In these experiments, tumor-bearing mice were treated with/without melanin; however, unfortunately, we did not observe differences between these two groups. Therefore in this study we omit the control group 5. In order that to increase the convincement for the study, we have added this explanation in the text, page 5, lines 127-129.       

Melanin nanoparticles are considered as iron chelation agents and can destroy cancer cells by iron starvation (10.1007/s10856-018-6190-x). Therefore, the combination of X-ray radiation and Melanin nanoparticles may consider as both radiotherapy and targeted drug therapy for cancer treatment, and the Melanin nanoparticles may not necessarily reduce the side effect of radiotherapy but they may directly kill cancer cells. The authors need to discuss this issue.

è Thank you so much for your nice recommendation. Beside the ability of melanin as a radioprotector, several previus studies reported that melanin could be used as a target molecule for cancer treatment, especially for melanoma treatment [1-4]. Because, melanin is synthesized and released by melanoma cells (and, of course melanocytes), therefore, it could be used as a target molecule for photothermal therapy and/or radioisotope therapy melanoma treatment rather than for treatment of other cancers [1-4]. In this study, we used lung cancer cells for the experiments; therefore, it was hard to see the impressive effect of melanin as an anticancer agent.

è We have discussed about this issue in the text, page 15, lines: 329-335.  

In addition, the novelty of the work is not clear. Is it the effect of Melanin nanoparticle on spleen or is the way that they have synthesized these nanoparticles? They need to elaborate on this matter in the introduction part.

è In this study, as you mentioned, the way we synthesized nanomelanin and applied this type of nanomelanin as a radioprotector is a novelty; And, another novelty of this study is the fact that we showed the activation of immunocells in spleens with nanomelanin treatment and more impressively, we also suggested that there may be a synergistic effect of X-ray and nanomelanin in promoting the population of T-lymphocytes and Dendritic cells.

è Following your comment, we have discussed more detail about this matter in the text, Page 16, lines: 364-368 and lines: 397-403

In Conclusion, the authors have mentioned: “In this study, we for the first time, prepared and characterized nanomelanin and investigated its ability in protecting healthy tissues of tumor-bearing mice from attack of X-ray during radiotherapy”. However, Rageh et al. in 2014 had already evaluated the effect of Melanin nanoparticles on spleen (10.1007/s11010-014-2232-y).

è In this study, we used different way to form the nanomelanin therefore we declared for the first time. However, as your comment, to avoid the misunderstanding, we have changed our statement in the conclusion part, page 17, line 414.  

Furthermore, the authors have only used a single concentration (40 mg/kg) of Melanin nanoparticles as well as a single dose of radiation (6.0 Gy) for their in vivo studies. Why have they selected this specific concentration and radiation dose? It would be more desirable to check the effect of various concentrations of melanin nanoparticles as well as radiation intensity and exposure time.

è Before approving the concentration of nanomelanin at 40 mg/Kg for radioprotecting experiment, toxicity of the nanomelanin had been tested. Mice were injected with two shots (2-day interval) of 1mL nanomelanin at different concentrations of 5, 10, 20, 40, 60, and 80 mg/Kg. We found that, at the maximum concentration of 60 mg/Kg, nanomelanin did not cause any defect or strange symptom of mice. However, at high concentrations of 80 mg/Kg, although mice were still alive they somehow lost their appetite and decreased their action. In addition, previous study also reported that nanomelanin at the concentration of 50 mg/Kg was safe for mice [5, 6]. Basing on these results, we decided to choose a safety dose of 40 mg/Kg (but high enough) for the radioprotecting experiments.

è In general, high doses of X-ray will be used to treat cancer on mice. We also tested toxicity of X-ray on mice. Mice were irradiated with different doses of 4, 5, 6, and 7 Gy with the high dose rate of 1 Gy/min. It showed that at the doses of 4, 5, and 6 Gy, X-ray did not caused any death of mice after 20 days post  irradiation; however, the dose of 7 Gy caused some deaths of mice after 7 days post irradiation. This result is consistent with results reported in previous studies [7-10]. Basing on this, we selected the dose of X-ray for further experiments.

è We also added these explanations in the text, page 8-9, lines: 204-216 and added references in the Reference part.    

 Minor issues:

1)      It’s better to add the size and concentration of nanoparticle in the abstract.

è Yes, we have added the information of concentration and size of the nanoparticles on the abstract part, page 2, lines: 25-29

2)      Figure 1 A-B needs scale bars.

è Yes, we have added the scale bars for the Figure 1A-B.

3)      The titles of vertical axes in Figure 4 B-C & Figure 5B are missing (there is only the unit of the axis).

è Yes, we have added the vertical axes in the figure 4B-C and Figure 5B.

4)      The authors have mentioned that on day 15 they sacrificed the mice. So, what are D18, D21 & D25 on Figure 4 B-C?

à We scarified all mice after 15 days post irradiation. And the irradiation time is at the day 10 when the tumor size reached to the desired volume (around 500 mm3). That means the total period for the experiment on mice is 25 days.

5)    There are several acronyms that have not been defined for the first time such as:

- Line 29: IL-2 (which is defined in Line 236), TNF-α

- Line 67: DMSO

- Line 100: PCR

- Line 104: RT

- Line 114: PBS

Also, some acronyms such as SEM and FTIR have been defined several times!

à Thank you so much. We have followed your comments to carefully check and correct all the mistakes in the texts.

6)      There are several syntax and grammatical errors, including, but not limited to, Lines 224, 236, 243, 370. The authors need to proofread the manuscript carefully.

è  Thank you so much. We have followed your comments to carefully check and correct all the mistakes in the texts.

è Actually, before submitting our manuscript to Materials, we had sent it to an English Editing Center (https://www.editage.com) for english checking and editing.

REFERENCES

1.      Jiang Q, Liu Y, Guo R, Yao X, Sung S, Pang Z, Yang W. Erythrocyte-cancer hybrid membrane-camouflaged melanin nanoparticles for enhancing photothermal therapy efficacy in tumors. Biomaterials. 192:292-308, 2019.

2.      Kim MKim HSKim MARyu HJeong HJLee CM. Thermohydrogel Containing Melanin for Photothermal Cancer Therapy. Macromol Biosci. 17(5). 2017.

3.      Joyal JLBarrett JAMarquis JC, Chen JHillier SMMaresca KPBoyd M, et al. Preclinical evaluation of an 131I-labeled benzamide for targeted radiotherapy of metastatic melanoma. Cancer Res. 70(10):4045-53, 2010.

4.      Dadachova EMoadel TSchweitzer ADBryan RAZhang TMints L, et al. Radiolabeled melanin-binding peptides are safe and effective in treatment of human pigmentedmelanoma in a mouse model of disease. Cancer Biother Radiopharm. 21(2):117-29, 2006.

5.       Rageh MMEl-Gebaly RHAbou-Shady HAmin DG. Melanin nanoparticles (MNPs) provide protection against whole-body ɣ-irradiation in mice viarestoration of hematopoietic tissues. Mol Cell Biochem. 399(1-2):59-69, 2015.

6.      Kunwar AAdhikary BJayakumar SBarik AChattopadhyay SRaghukumar SPriyadarsini KI. Melanin, a promising radioprotector: mechanisms of actions in a mice model. Toxicol Appl Pharmacol.264(2):202-211, 2012.

7.      Hayashi MHirai RIshihara YHoriguchi NEndoh DOkui T. Combined effects of treatment with trientine, a copper-chelating agent, and x-irradiation on tumor growth in transplantation model of a murine fibrosarcoma. J Vet Med Sci. 69(10):1039-1045, 2007.

8.      Ueno MInano HOnoda MMurase HIkota NKagiya TVAnzai K. Modification of mortality and tumorigenesis by tocopherol-mono-glucoside (TMG) administeredafter X irradiation in mice and rats. Radiat Res. 172(4):519-24, 2009.

9.      Anzai KUeno MMatsumoto KIkota NTakata J. Gamma-tocopherol-N,N-dimethylglycine ester as a potent post-irradiation mitigator against wholebody X-irradiation-induced bone marrow death in mice. J Radiat Res. 55(1):67-74, 2014.

10.  Abdullaev SMinkabirova GKarmanova EBruskov VGaziev A. Metformin prolongs survival rate in mice and causes increased excretion of cell-free DNA in the urine of X-irradiated rats. Mutat Res Genet Toxicol Environ Mutagen. 831:13-18, 2018.

Reviewer 2 Report

1. In Figure 4A: Please show the representative image of all groups.

2. In Figure 4C: Apparently IR and IR + ML group have similar tumor regression. Do you expect any better outcome in terms of tumor growth or ML helps only with radiation-induced damage?

3. In Figure 4: Did the authors optimise the dose of ML to 40 mg/kg? If yes, please share the data. I am curious to know whether how the effective dose was selected?

4. Blood work of mice groups also depicts a nil to modest improvement in the profile while comparing IR and IR+ ML groups. Error bars are so tight. How many mice were accounted in this experiment? Please explain.

5. In figure 5A, Please include all five spleen images.

5. Both Il-2 and TNF alpha are multifunctional cytokines and it is hard to use it an evidence to show  increased splenic weight and upregulated immune response. Moreover, gene expression in splenocytes can be used as a surrogate marker for their role on effector cells. Please either provide the serum levels of these cytokines or perform a cell based cytotoxicity assay to account for this phenotype. 

6. CD11c is a type I transmembrane protein that is expressed on monocytes, granulocytes, a subset of B cells, dendritic cells, and macrophages.  it cannot be used to show the dendritic cells only (activated or knaive is also debatable).

7.In Figure 8, please include representative images from all groups.

8. Again, please include IHC of all apoptotic markers with active caspase antibodies.

Author Response

Dear Reviewer,

First of all, we would like to thank Editors and Reviewers for their considerations and reasonable comments to our manuscript. We have followed all the comments to revise our manuscript by making clear several points, supplying additional data and correcting all the typo and grammar mistakes in the manuscript.

Now we would like to re-submit our revised-manuscript entitled “Nanomelanin potentially protects the spleen from radiotherapy-associated damage and enhances immunoactivity in tumor-bearing mice”, (manuscript ID: materials-503306to Materials of MDPI for your further considerations. In the revised manuscript, the revised parts are maked with red color.

In this letter, we also attach our responses to the comments of the Reviewer.

Thank you so much.

Best regards,

Corresponding author: Nguyen Dinh Thang

²  Response to the Reviewer 2

1.      In Figure 4A: Please show the representative image of all groups.

è Yes, we showed the representative image of all group in the Figure 4A

2.      In Figure 4C: Apparently IR and IR + ML group have similar tumor regression. Do you expect any better outcome in terms of tumor growth or ML helps only with radiation-induced damage?

è Beside the ability of melanin as a radioprotector, several previous studies reported that melanin could be used as a target molecule for cancer treatment, especially for melanoma treatment [1-4]. Because, melanin is synthesized and released by melanoma cells (and, of course melanocytes), therefore, it could be used as a target molecule for photothermal therapy and/or radioisotope therapy melanoma treatment rather than for treatment of other cancers [1-4]. In this study, we used lung cancer cells for the experiments; therefore, it was hard to see the impressive effect of melanin as an anticancer agent.

è We have also discussed about this issue in the text, page 15, lines: 329-335.

3.      In Figure 4: Did the authors optimise the dose of ML to 40 mg/kg? If yes, please share the data. I am curious to know whether how the effective dose was selected?

è Before approving the concentration of nanomelanin at 40 mg/Kg for radioprotecting experiment, toxicity of the nanomelanin had been tested. Mice were injected with two shots (2-day interval) of 1mL nanomelanin at different concentrations of 5, 10, 20, 40, 60, and 80 mg/Kg. We found that, at the maximum concentration of 60 mg/Kg, nanomelanin did not cause any defect or strange symptom of mice. However, at high concentrations of 80 mg/Kg, although mice were still alive they somehow lost their appetite and decreased their action. In addition, previous study also reported that nanomelanin at the concentration of 50 mg/Kg was safe for mice [5, 6]. Basing on these results, we decided to choose a safety dose of 40 mg/Kg (but high enough) for the radioprotecting experiments.

è In general, high doses of X-ray will be used to treat cancer on mice. We also tested toxicity of X-ray on mice. Mice were irradiated with different doses of 4, 5, 6, and 7 Gy with the high dose rate of 1 Gy/min. It showed that at the doses of 4, 5, and 6 Gy, X-ray did not caused any death of mice after 20 days post  irradiation; however, the dose of 7 Gy caused some deaths of mice after 7 days post irradiation. This result is consistent with results reported in previous studies [7-10]. Basing on this, we selected the dose of X-ray for further experiments.

è We also added these explanations in the text, page 8-9, lines: 204-216 and added references in the Reference part.       

4.      Blood work of mice groups also depicts a nil to modest improvement in the profile while comparing IR and IR+ ML groups. Error bars are so tight. How many mice were accounted in this experiment? Please explain.

è We had five mice for each group. After scarifying of mice, blood samples were sent to the clinical center for hematological analysis of several parameters, such as WBC, RBC, HGB, HCT, PLT and LYM. After analyzing, the center sent us the data and we statistically analyzed to get the results. The error bars came from the data processing using statistical analysis tool.       

5.      In figure 5A, Please include all five spleen images.

è We actually have five mice for each group at the scarifying time, however, at the beginning point, our technicians forgot the protocol and quickly cut some spleens into several pieces for different purposes (sectioning, RNA isolating, protein extracting,...) just after weighing. Therefore, we have only four uncut spleens for each group and we have replaced the figure 5A (with three spleen samples for each group) with the new one (with four spleen samples for each group).  

6.      Both Il-2 and TNF alpha are multifunctional cytokines and it is hard to use it an evidence to show increased splenic weight and upregulated immune response. Moreover, gene expression in splenocytes can be used as a surrogate marker for their role on effector cells. Please either provide the serum levels of these cytokines or perform a cell based cytotoxicity assay to account for this phenotype.

è Thank you so much for your nice recommendations. We know that both IL-2 and TNF-anpha are multifunctional cytokines and totally agree that supplying the results of IL-2 and TNF-anpha levels in serum and/or results of splenic cell based cytotoxic assay will provide better distinct role of these cytokines in increasing the splenic spleen and upregulating the immune response. However, unfortunately, at present time, we are sorry that we do not have serum samples for further experiments. Because after scarifying the mice, blood sample (serum included) had been sent to the clinical center for hematological analysis (WBC/RBC/PLT/HGB/HCT/LYM) and presented in the table 2. In the other hand, the role of IL-2 and TNF-anpha in increasing the splenic spleen and upregulating the immune response are well known. Therefore, we think that, the current evident are somehow enough to prove the role of the two genes in this study.   

7.      CD11c is a type I transmembrane protein that is expressed on monocytes, granulocytes, a subset of B cells, dendritic cells, and macrophages.  it cannot be used to show the dendritic cells only (activated or knaive is also debatable).

è There are many subsets of Dendritic cells (DCs) and different subset of DCs has its own distinct panel of markers on the cell surface. However, all DCs are CD11c-positive cells. Therefore CD11c is mainly accepted and widely used as the first priority as a marker when scientists want to determine and gate the DCs population. Although in the spleen, beside the DCs, there are several other cells such as monocytes, macrophases, and neutrophils; however, in normal conditions, monocytes, macrophases, and neutrophils are CD11c-negative cells [11-14]. In some circumstances, especially under infection with bacteria, a few amounts of monocytes, macrophases, and lymphocytes with CD11c-positive may appear in the spleens of mice; however, the percentages of these CD11c-positive cells are much smaller than that of DCs [13-15]. In addition, in this study, the mice had not been infected by bacteria; therefore the population of CD11c-positive cells but not DCs in the spleens of mice should be very low.

è We also added this explanation in the text, page 12, lines: 283-293

8.      In Figure 8, please include representative images from all groups.

è Yes, we have added the representative images for all groups in the Figure 8.

9.      Again, please include IHC of all apoptotic markers with active caspase antibodies.

è Thank you for your comment. Actually, we are also planning to perform several experiments to examine the role of nanomelanin in cancer treatment; And apoptosis is one of the most important pathways we should focus on. However, in this study, we have mainly concentrated on investigating the role of nanomelanin as a radioprotector and its ability in inducing immunoactivity rather than its role in cancer treatment. Therefore, please let us set aside these experiments for the next study related to the theme “role of nanomelanin in cancer treatment”.  

REFERENCES

1.      Jiang Q, Liu Y, Guo R, Yao X, Sung S, Pang Z, Yang W. Erythrocyte-cancer hybrid membrane-camouflaged melanin nanoparticles for enhancing photothermal therapy efficacy in tumors. Biomaterials. 192:292-308, 2019.

2.      Kim MKim HSKim MARyu HJeong HJLee CM. Thermohydrogel Containing Melanin for Photothermal Cancer Therapy. Macromol Biosci. 17(5). 2017.

3.      Joyal JLBarrett JAMarquis JC, Chen JHillier SMMaresca KPBoyd M, et al. Preclinical evaluation of an 131I-labeled benzamide for targeted radiotherapy of metastatic melanoma. Cancer Res. 70(10):4045-53, 2010.

4.      Dadachova EMoadel TSchweitzer ADBryan RAZhang TMints L, et al. Radiolabeled melanin-binding peptides are safe and effective in treatment of human pigmentedmelanoma in a mouse model of disease. Cancer Biother Radiopharm. 21(2):117-29, 2006.

5.       Rageh MMEl-Gebaly RHAbou-Shady HAmin DG. Melanin nanoparticles (MNPs) provide protection against whole-body ɣ-irradiation in mice viarestoration of hematopoietic tissues. Mol Cell Biochem. 399(1-2):59-69, 2015.

6.      Kunwar AAdhikary BJayakumar SBarik AChattopadhyay SRaghukumar SPriyadarsini KI. Melanin, a promising radioprotector: mechanisms of actions in a mice model. Toxicol Appl Pharmacol.264(2):202-211, 2012.

7.      Hayashi MHirai RIshihara YHoriguchi NEndoh DOkui T. Combined effects of treatment with trientine, a copper-chelating agent, and x-irradiation on tumor growth in transplantation model of a murine fibrosarcoma. J Vet Med Sci. 69(10):1039-1045, 2007.

8.      Ueno MInano HOnoda MMurase HIkota NKagiya TVAnzai K. Modification of mortality and tumorigenesis by tocopherol-mono-glucoside (TMG) administeredafter X irradiation in mice and rats. Radiat Res. 172(4):519-24, 2009.

9.      Anzai KUeno MMatsumoto KIkota NTakata J. Gamma-tocopherol-N,N-dimethylglycine ester as a potent post-irradiation mitigator against wholebody X-irradiation-induced bone marrow death in mice. J Radiat Res. 55(1):67-74, 2014.

10.  Abdullaev SMinkabirova GKarmanova EBruskov VGaziev A. Metformin prolongs survival rate in mice and causes increased excretion of cell-free DNA in the urine of X-irradiated rats. Mutat Res Genet Toxicol Environ Mutagen. 831:13-18, 2018.

11.  Ying Y. Hey, Helen C. O’Neill . Murine spleen contains a diversity of myeloid and dendritic cells distinct in antigen presenting function. J. Cell. Mol. Med. 16(11):2611-2619, 2012.

12.  Shalin H NaikDonald MetcalfAnnemarie van NieuwenhuijzeIan WicksLi WuMeredith O'Keeffe, Ken Shortman. Intrasplenic steady-state dendritic cell precursors that are distinct from monocytes. Nature Immunology. 7:663–671, 2006.

13.  Mildner AJung S. Development and function of dendritic cell subsets. Immunity. 40:642-56, 2014.

14.  Kristin L Griffiths, Jonathan KH Tan, Helen C O’Neill . Characterization of the effect of LPS on dendritic cell subset discrimination in spleen. J. Cell. Mol. Med. 18: 9, 2014.

15.  Arnold IC, Mathisen S,  Schulthess J, Danne C, Hegazy AN and Powrie F. CD11c+ monocyte/macrophages promote chronic Helicobacter hepaticus-induced intestinal inflammation through the production of IL-23. Mucosal Immunol. 9(2):352-363, 2016.

Round 2

Reviewer 1 Report

The authors have addressed all my comments and the revised version can be accepted now.

Reviewer 2 Report

The authors categorically answered all the concerns. I support this manuscript for publication.